# Green Activated Magnetic Graphitic Carbon Oxide and Its Application for Hazardous Water Pollutants Removal

**Lakshmi Prasanna Lingamdinne**[ID]**, Jong-Soo Choi, Yu-Lim Choi, Jae-Kyu Yang \*,
Janardhan Reddy Koduru \***[ID] **and Yoon-Young Chang \***

Department of Environmental Engineering, Kwangwoon University, Seoul 01897, Korea
\* Correspondence: jkyang@kw.ac.kr (J.-K.Y.); reddyjchem@gmail.com (J.R.K.); yychang@kw.ac.kr (Y.-Y.C.);
Tel.: +82-02-940-5496 (J.R.K.)

**Abstract:** Graphitic carbon oxide (GCO) and magnetic graphitic carbon oxide (MGCO) were prepared from sugar via optimized green activation by employing ozone oxidation, and applied to wastewater treatment. The maximal oxidation and adsorption yield of pollutants were achieved at pH 2.0–4.0, which is the optimized pH for ozone oxidation of GC to generate GCO. As-prepared GCO and MGCO were characterized using X-ray, infrared, and microscopic techniques. The MGCO has enough saturation magnetization ($M_S$) of 41.38 emu g$^{-1}$ for separation of the sorbent from the reaction medium by applying an external magnetic field. Batch adsorption of radioactive and heavy metals (Th(IV), Pb(II)), and a dye (methylene blue (MB)) using GCO and MGCO was evaluated by varying the adsorbent dose, equilibrium pH, contact time, initial metal and dye concentrations, and kinetics and isotherms. Adsorption kinetics and isotherm studies indicated that Th(IV), Pb(II), and MB adsorption were best described by pseudo-second-order kinetics and Langmuir isotherm with $R^2$ (correlation coefficient) > 0.99, respectively. The maximum adsorption capacities for Th(IV), Pb(II), and MB were 52.63, 47.39, and 111.12 mg g$^{-1}$ on GCO and 76.02, 71.94, and 76.92 mg g$^{-1}$ on MGCO. GCO and MGCO are prospectively effective and low-cost adsorbents for ion removal in wastewater treatment. As prepared MGCO can be reused up to three cycles for Th(IV), Pb(II), and MB. This work provides fundamental information about the equilibrium adsorption isotherms and mechanisms for Th(IV), Pb(II), and MB on GCO and MGCO.

**Keywords:** metals; adsorption; kinetics; graphitic carbon; ozonation; magnetic graphitic carbon oxide

---

## 1. Introduction

Rapid industrialization has led to the release of harmful pollutants into various environments such as air, water, and soil. Water pollution is the primary universal environmental issue, and the release of harmful pollutants into water bodies is well above the standard limits [1,2]. Radionuclides (e.g., Th(IV)), heavy metals (e.g., Pb(II)), and dyes (e.g., MB) are the major water pollutants [3,4]. The use of metals and metalloids and organic dyes in various anthropogenic activities, including nuclear power plants, mining operations, and industries for various applications, is the major source of water pollution. Continuous exposure to these toxic chemicals (heavy metals, radionuclides, and organic dyes) poses a high health risk for human beings [1]. For example, exposure to Th(IV) leads to effects on the liver, spleen, and marrow [5], and trace levels of Pb(II) adversely affect the kidneys, nervous system, reproductive system, liver, and brain, resulting in sickness or death [6,7]. The permissible level of lead in drinking water is 0.05 mg·L$^{-1}$ [8]. The primary source of dyes (e.g., MB) is textile industry wastewater. Most of these wastewaters contain toxic compounds and different types of

carcinogenic dyes. They are toxic to human beings, fish species, and microorganisms [9]. Industries and human activities emit large quantities of dangerous metals and dyes into water bodies, necessitating treatment of contaminated water prior to release into the environment. There are several methods available for the treatment of water contaminated with dyes and metal ions including membrane separation, photocatalytic approaches, degradation using various microorganisms, chemical oxidation, filtration, adsorption, and coagulation [10,11]. Among these techniques, adsorption is the most facile for removing metals and dyes from aqueous media, where the pollutants are transferred from the aqueous medium to a solid-phase adsorbent. Moreover, adsorption techniques offer low installation costs with high efficiency and are affordable to developing nations.

Nowadays, graphene oxide (GO) and its descendants are extensively applied in water restoration due to its high chemical stability having the large surface area and high activity. Generally, graphene oxide is synthesized by chemical methods such as the Hummer's method [12], where strong acids are used in the oxidation process to generate effective adsorbents for water contaminants [13–20]. Such materials have higher adsorption capacity than activated carbon at an equivalent mass of carbon. Even though oxidized activated carbon materials show high efficiency, the oxidation or activation process using strong and toxic acids and high energy, and the production of hazardous by-products during their synthesis, are not eco-friendly and pose an environmental risk.

One recent study reported the preparation of cost-effective graphitic carbon from edible sugar by pyrolysis for the restoration of water from the organic explosives and dyes [3,21]. Recently, the oxidation of graphene using ozone and O ions was reported [22–26]. These studies prompted us to explore the oxidation of graphitic carbon prepared from sugar by ozonation via a green approach using ozone generated from oxygen gas, where the ozone molecules react with graphitic carbon to produce O-containing functional groups at the graphitic carbon. The oxidized graphitic carbon possesses –OH, –C=O, and –COOH surface functional groups that are favorable for the adsorptive removal of contaminants from wastewater. However, the composites are nano-sized, not magnetically separable, and complicated to separate from aqueous solution. Therefore, we further developed an economic magnetic graphene oxide-based granular material with high sorption capacity that is magnetically separable. Specifically, we attempted to prepare magnetically separable granular composite materials through impregnation of $Fe^{2+}/Fe^{3+}$ into graphitic carbon oxide in order to produce magnetic pseudo-graphitic carbon oxide composites. The main objectives of this study were (i) to prepare green and cost-effective graphitic carbon oxide and magnetic graphitic carbon oxide, (ii) to physically and chemically characterize the prepared composites, and (iii) to evaluate the characteristics of the prepared materials for adsorption of water pollutants, including organic pollutants, heavy metals, and radionuclides.

## 2. Materials and Methods

### 2.1. Graphitic Carbon Preparation

Graphitic carbon was prepared from sugar as described in our previous papers [3,21]. Briefly, sugar was dehydrated at ~120 °C to produce caramel. The produced caramel as heated in a temperature-programmed furnace with different residence times under $N_2$ atmosphere. The heating program was as follows: (i) in the first 30 min, the temperature was increased from room temperature to 100 °C; (ii) the temperature was increased to 200 °C in the next 30 min, and (iii) was held it for 1 h, (iv) then increased to 750 °C over the next 1 h and held for 3 h. The furnace was then cooled to room temperature at 1 °C·min$^{-1}$ over 12 h. The material was collected and ground to a powder. After several experiments, it was found that the material showed a large surface area and pore volume at 750 ± 5 °C. Hence, 750 °C was chosen as an optimal pyrolysis temperature for making graphitic carbon from edible sugar. The resultant product is denoted as graphitic carbon (GC) here onwards.

## 2.2. Graphitic Carbon Oxide (GCO) Preparation

Graphitic carbon oxide (GCO) was prepared as reported in our previous paper [27]; the details are shown in Figure 1. Briefly, the above-prepared GC dispersed in water and then the solution pH regulated to 4 by adding dilute HCl (0.1 mol·L$^{-1}$) solution. The solution was purged with O$_3$ (5 g·h$^{-1}$) gas that was produced through an ozone generator. After the ozonation process, the oxidized graphitic carbon was filtered using a 0.45 µm filter paper and washed with water and 50% ethanol. The washed compound was dried at 80 °C for 8 h and is denoted as graphitic carbon oxide (GCO).

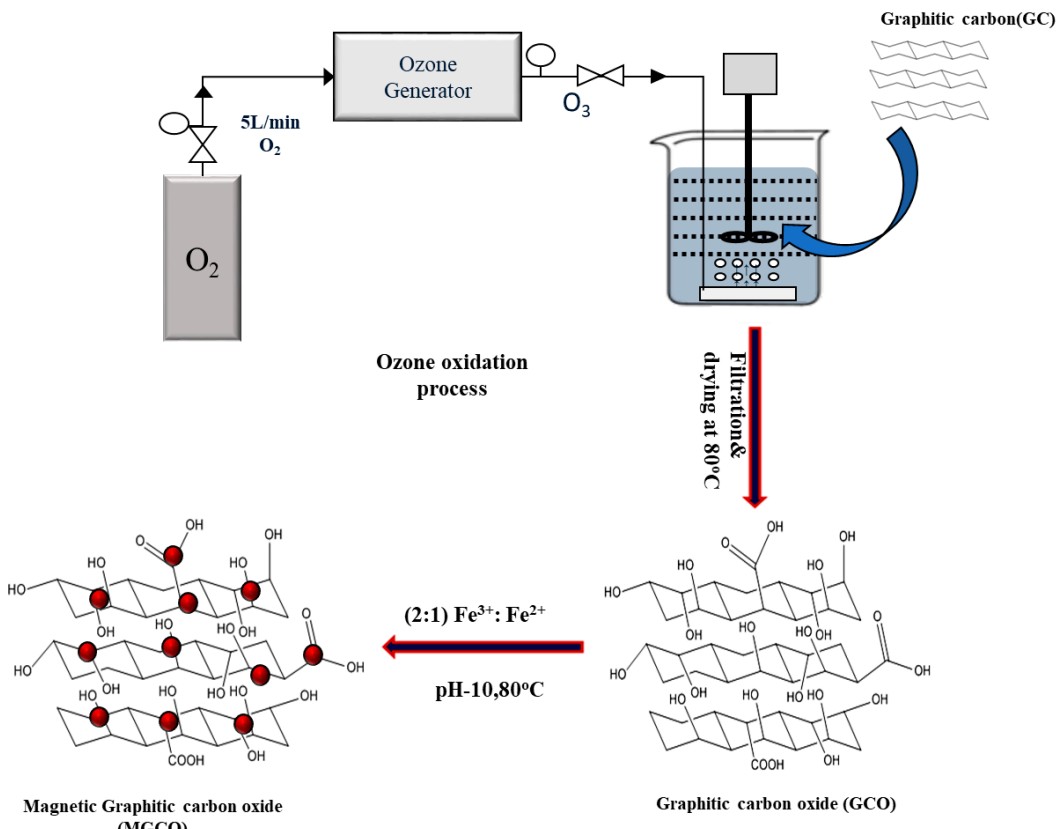

**Figure 1.** Schematic presentation of green oxidation of graphitic carbon using O$_3$ produced with the aid of ozone generator for the production of graphitic carbon oxide (GCO) and magnetic graphitic carbon oxide (MGCO).

## 2.3. Magnetic Graphitic Carbon Oxide (MGCO) Preparation

As shown in Figure 1, to the above prepared 1 g of GCO has been dispersed in 500 mL water by 30 min sonication with aid of ultra sonicator, followed by slow addition of a 2:1 Fe$^{3+}$/Fe$^{2+}$ solution under stirring and N$_2$ atmosphere. The solution pH was adjusted to 10 using 27% ammonia solution with continuous stirring. The solution was warmed up to 80 °C and stirred for 45 min. After completing the reaction, the solution was cooled to room temperature, filtered, and washed with DI water to achieve neutral pH. The material has been collected with aid of external magnet, washed with water and dried at 80 °C in a vacuum oven for 8 h. The physicochemical properties of the prepared material was confirmed by X-ray diffraction (XRD), X-ray photoelectron spectroscopy (XPS), Fourier-transform infrared (FT-IR), Raman, scanning electron microscopy-energy dispersive X-ray (SEM-EDX), thermal gravimetric analysis (TGA), and transmission electron microscopy (TEM) and is denoted as magnetic graphitic carbon oxide (MGCO).

### 2.4. The Adsorptive Procedure of GCO and MGCO for Removal of Pollutants

To evaluate the surface adsorptive performance of prepared GCO and MGCO for Th(IV), Pb(II), and MB adsorptive removal, batch studies were carried out in a falcon tube (50 mL). The reaction pollutant solution (50 mL) was prepared from a stock solution (1 mg·L$^{-1}$) by dilution with DI water. Here, 0.1 mol·L$^{-1}$ HCl or NaOH were used to adjusting the solution pH required. The required amount of GCO or MGCO was added to the solution, and the series of samples were rotated at 150 rpm. The required portion of the solution was collected and immediately filtered through 0.45 μm syringe filters. The filtrate contains Pb(II) and Th(IV) concentration was measured by using inductively coupled plasma-optical emission spectroscopy (Optima 2100 DV, ICP-OES, Perkin-Elmer, Waltham, MA, USA) to estimate the adsorptive amount of metal ions. While the filtrate contains MB was measured by using ultraviolet-visible spectrophotometry (UV 1601 PC, Shimadzu, Kyoto, Japan). The adsorption isotherms were obtained with different initial concentrations (2.0−30 mg·L$^{-1}$) of Th(IV), Pb(II), and MB. All experiments were performed at least in duplicate at ambient temperature (25 ± 2 °C). All data reported are the mean±standard deviation (SD). Moreover, a reagent blank was evaluated to determine the adsorption associated with the instruments used in the adsorption experiment, which was negligible. A reusability test was performed to check the ability for reuse and regeneration of the prepared materials (GCO and MGCO). For this purpose, the pollutant-loaded GCO and MGCO were collected and a desorption test was performed with suitable desorbing agents, including 0.1 mol·L$^{-1}$ HNO$_3$ (for Pb(II) and Th(IV)) and pure acetonitrile (for MB).

The adsorption kinetics was evaluated by applying the pseudo-first-order (PFO) and pseudo-second-order (PSO) kinetics models to the pollutant adsorption kinetics data. PFO assumes that the adsorption rate is proportional to the difference between the adsorption capacities at equilibrium ($q_e$, mg g$^{-1}$) and time $t$ (min) ($q_t$, mg g$^{-1}$). PSO considers that the process follows chemical adsorption involving the adsorbent and adsorbate. The two kinetic model equations are:

$$\text{PFO kinetic equation}: \ \log(q_e - q_t) = \log(q_e) - \left(\frac{K_1}{2.303}\right)t \tag{1}$$

$$\text{PSO kinetic equation}: \ \frac{t}{q_t} = \left(\frac{1}{K_2\, q_e^2}\right) + \left(\frac{1}{q_e}\right)t \tag{2}$$

here k$_1$ and k$_2$ are the kinetics rate constants of PFO and PSO, respectively. The mechanism of isothermal adsorption was evaluated with the often-used isotherm models (Langmuir, Freundlich, and Temkin models) and are expressed as follows:

$$\text{Langmuir isotherm equation}: \ q_e = q_{max} \cdot K_L \cdot C_e / 1 + k_L C_e \tag{3}$$

$$\text{Freundlich isotherm equation}: \ q_e = K_F \cdot C_e^{1/n} \tag{4}$$

$$\text{Temkin isotherm equation}: \ q_e = B \log K_T + B \log C_e \tag{5}$$

here $C_e$, $q_{max}$ (mg·g$^{-1}$), K$_L$ (L·mg$^{-1}$), and K$_F$ (mg·g$^{-1}$(L·mg$^{-1}$) 1/n) are the equilibrium concentration of the pollutant molecule, the maximum adsorption capacity, Langmuir adsorption equilibrium constant, and Freundlich isotherm model constant, respectively. In Equation (4), $n$ indicates the degree of non-linearity between the adsorbate concentration and adsorption. In Equation (5), $B$ (g·L$^{-1}$) and log$K_T$ (L·mg$^{-1}$) indicate the equilibrium constants of the Temkin isotherm.

## 3. Results and Discussion

### 3.1. Optimization of the Ozonation Process with Solution pH

Due to the short half-life of ozone, a certain period is required for the oxidation process [28]. Herein, we determined the optimal pH for increasing the half-life of ozone. The adsorption performance of GCO prepared from ozonation of GC under different pH conditions was evaluated using three

pollutants (Th(IV), Pb(II), and MB), as shown in Figure 2. Higher adsorptive removal was observed at lower (or acidic) pH (2 or 4). However, the adsorptive removal of the pollutants decreases to a lower range by the solution pH increases from 6 to 12 because the saturated ozone concentration increased at lower pH in aqueous solution. However, at higher pH, ozone is rapidly decomposed due to the presence of hydroxyl ions [29,30]. Based on those results, pH 4 was chosen as the optimum pH for the ozonation process for conversion of GC to GCO. The greater number of oxygen-containing functional groups has been observed in GCO prepared by ozonation carried out at lower pH as compared to that ozonation carried out at higher pH from FT−IR (it is not reported here). Due to the high oxygen-containing functional groups of the GCO prepared at lower pH shows higher adsorption. Further, the ozonation time effect on GC oxidation was studied and the results found that the ozonation time was an insignificant effect on GC oxidation. From this optimization studies, pH 4.0, temperature 25 °C, time 1 h, and a disc-type bubbler were used in the ozonation process as an optimized condition for the oxidation of GC to GCO.

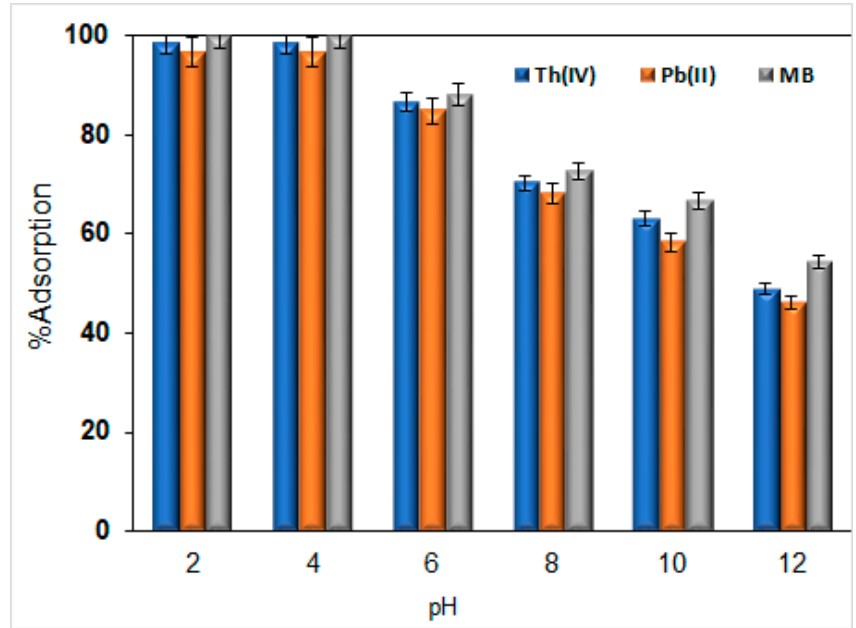

**Figure 2.** Adsorption performance of GCO (1.0 g·L$^{-1}$) for three pollutants (Th(IV), Pb(II), and MB = 10 mg·L$^{-1}$) prepared from ozonation of GC at different pH (2, 4, 6, 8, 10, and 12), temperature 25 °C, and time 1 h.

### 3.2. Experimental Parameters for Adsorptive Removal of Pollutants by MGCO and GCO

The percentage removal of Th(IV), Pb(II), and MB with GCO and MGCO as a function of time is plotted in Figure 3a,b. The results show that the adsorptive removal of Th(IV) was rapid and maximum removal of ~97% was achieved within 5 min of equilibrium for MGCO, followed by a plateau at a longer time. In the case of GCO, there was a slow increase, and the system gets equilibrium at 30 min with 78% adsorption. While Pb(II) adsorption increased slowly and reached the maximum at 120 min with 76.3 and 95% removal for GCO and MGCO, respectively. However, in the case of MB, an interesting phenomenon was observed; that is, the equilibrium time (45 min) and maximum adsorption removal (80%) decreased for MGCO relative to that of GCO (at 60 min equilibrium time, more than 99% maximum removal was observed). Figure 3a,b shows that the maximum adsorption removal of both metal ions was higher for MGCO than that for GCO, whereas the trend was opposite for MB adsorption. This may be because adsorption of the MB ions involved ion exchange by interaction with the surface functional groups and with the magnetic iron oxide particles.

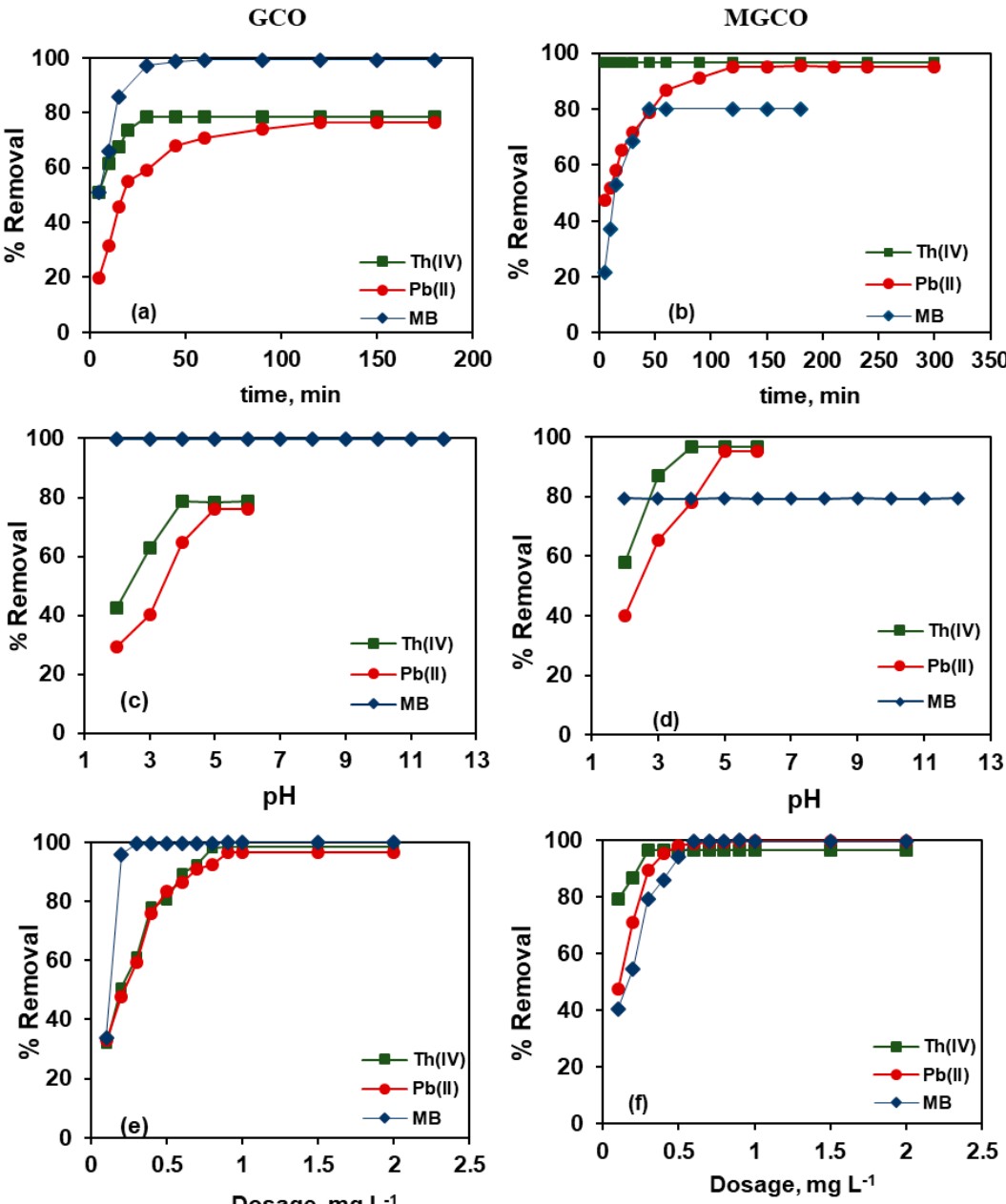

**Figure 3.** Studies of (**a**,**b**) contact time, (**c**,**d**) pH, and (**e**,**f**) GCO and MGCO dosage effects on adsorptive removal of Th(IV), Pb(II), and MB from aqueous medium by GCO and MGCO. (Experimental conditions: pH: 5.0, Amount of GCO and MGCO: 0.3 g·L$^{-1}$, equilibrium time: 30 min for Th(IV) on GCO and MGCO, and MB on GCO; 45 min for MB on MGCO, 90 min for Pb(II) on GCO and MGCO, temperature: 298 K, Th(IV), Pb(II), and MB initial concentration: 10 mg·L$^{-1}$.

The fitted PFO and PSO kinetic models and calculated kinetics parameters are presented in Table 1 and Figure S1.

**Table 1.** Kinetics parameters GCO and MGCO (0.3 g·L$^{-1}$) nanocomposites for the Th(IV), Pb(II), and MB (10 mg·L$^{-1}$) adsorptive removal at pH 5.0. The reported values are mean ± Standard deviation.

| Name of Adsorbent | Pollutant | $q_e$, Th, mg·g$^{-1}$ | PFO | | | PSO | | |
|---|---|---|---|---|---|---|---|---|
| | | | $q_{e, Cal.}$ mg·g$^{-1}$ | $K_1$ | $R^2$ | $q_{e, Cal.}$ mg·g$^{-1}$ | $K_2$ | $R^2$ |
| GCO | Th(IV) | 19.05 | 23.47 | 0.32 | 0.818 | 18.38 | 0.031 | 0.999 |
| | Pb(II) | 18.95 | 11.035 | 0.034 | 0.925 | 19.01 | 0.005 | 0.998 |
| | MB | 38.85 | 49.97 | 0.15 | 0.910 | 37.34 | 0.055 | 0.999 |
| MGCO | Pb(II) | 23.05 | 13.00 | 0.037 | 0.912 | 23.31 | 0.004 | 0.999 |
| | MB | 32.56 | 25.62 | 0.149 | 0.845 | 33.34 | 0.047 | 1 |

From theoretical adsorption capacity, $q_{e, Th.}$ and calculated adsorption capacity, $q_{e, Cal.}$ and the correlation coefficient ($R^2$), the PSO model adequately describes the kinetics of Th(IV), Pb(II), and MB adsorption on GCO and MGCO. Thus, it is concluded that the adsorption of Th(IV), Pb(II), and MB on GCO and MGCO was a rate-limiting diffusion reaction.

The removal percentages of GCO and MGCO for Th(IV), Pb(II), and MB at different initial pH of the solution are illustrated in Figure 3c,d. The figure shows that the removal percentage of Th(IV) increased as the initial pH of the solution increased from 2 to 4 and reached a maximum at pH 4−6. While reached maximum removal at pH 5 in the case of Pb(II). Thereafter, the removal plateaued as the initial pH of the solution increased from 4.0 to 6.0 (for Th(IV)) and 5.0 to 6.0 (for Pb(II)). The trend for MB was quite different from that of the metal ions, wherein the adsorptive removal was not affected significantly by the solution initial pH and was constant at all solution pH conditions (2 to 12) with both GCO and MGCO. However, the removal capacity of both adsorbents differed, and interestingly, GCO showed higher adsorption than MGCO. This means that the magnetization of GCO decreased the MB adsorption. Generally, Pb(II) removal proceeds well between pH 4.0 and 6.0 in aqueous solution. Above pH 6, the metal tends to precipitate out of solution as the hydroxide. In the case of MB, the adsorptive removal was maximal and constant at all pH (pH 2–12). Hence, for further studies, pH 5.0 was used for the three target pollutants.

The dosage effect of GCO and MGCO for the adsorptive removal of Th(IV), Pb(II), and MB is shown in Figure 3e,f. The maximum percentage removal of Th(IV), Pb(II), and MB were obtained with adsorbent concentrations of 0.8, 0.9, and 0.3 g·L$^{-1}$ for GCO and 0.3, 0.5, and 0.6 g·L$^{-1}$ for MGCO. The removal curve reached a plateau when the adsorbent dosage was further increased. However, the observed $q_e$ decreased with the adsorbent dosage increased. Typically, as the adsorbent dosage increases, the molecular interactions between the adsorbent molecules increases, leading to a decrease in the number of sites for a specific area, which may lead to reduced adsorption capacity ($Q_e$). Interestingly, the concentration of MGCO required for maximizing the removal of MB was higher than that of GCO. In the case of the other two metal pollutants, the amount of MGCO required for maximizing the adsorptive removal was lower than that of GCO.

Industrial wastewater and natural water bodies contain cationic ions such as $Na^+$, $K^+$, $Ca^{2+}$, and $Mg^{2+}$. Hence, the effect of these ions on the uptake of Th(IV), Pb(II), and MB was investigated as illustrated in Figure S2. The adsorption studies were performed with ion concentrations of 0.005 to 0.5 mg·L$^{-1}$. The results indicate that the capacity of GCO and MGCO to adsorb Th(IV), Pb(II), and MB ions was not influenced when the cation concentration increased.

*3.3. Adsorption Isotherms*

After determining the optimum conditions for maximum adsorptive removal of the three contaminants from the above studies, the adsorption of different concentrations of contaminants was investigated and the resultant isotherms are shown in Figure 4. The data were fitted to the well-known Langmuir, Freundlich, and Temkin isotherm models to understand the adsorption process (Figure S3). The obtained isotherm parameters are reported in Table 2.

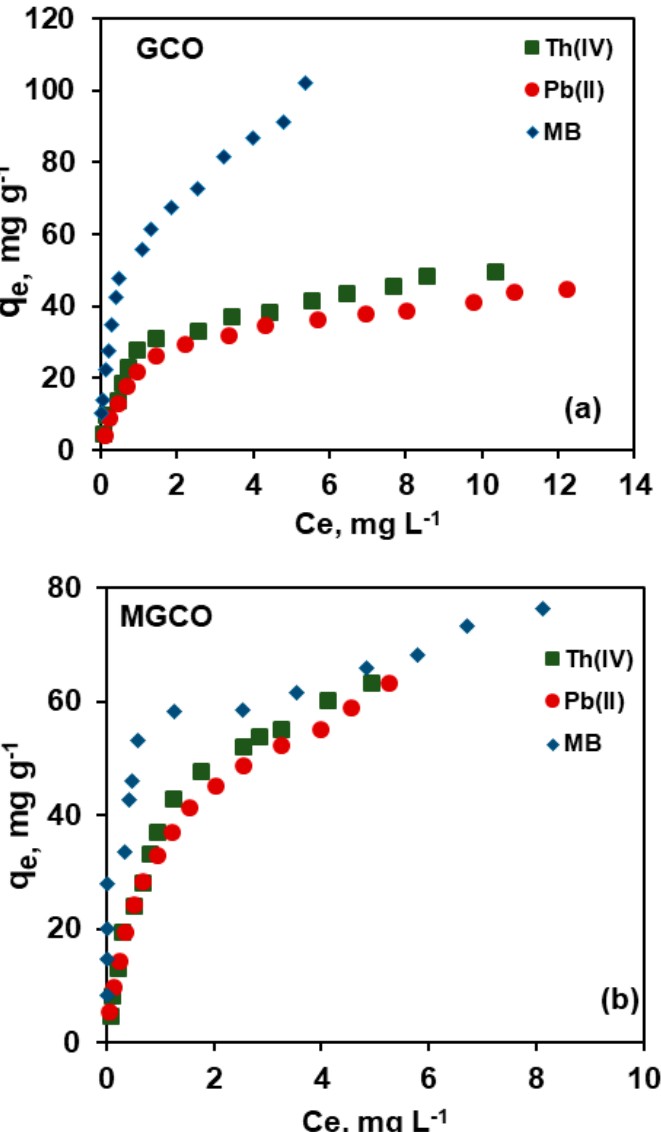

**Figure 4.** Equilibrium isotherm curves for Th(IV), Pb(II), and MB (2 to 30 mg·L$^{-1}$) (**a**) on GCO (0.3 g.L$^{-1}$) and (**b**) MGCO (0.3 g.L$^{-1}$). Solution initial pH: 5.0 for three pollutants; equilibrium time: 30 min for Th(IV) on GCO and MGCO and MB on GCO; 45 min for MB on MGCO; 90 min for Pb(II) on GCO and MGCO; temperature: 298 K were used in the experiment.

**Table 2.** Adsorption isotherm parameters for Th(IV), Pb(II), and MB (2 to 30 mg·L$^{-1}$) on GCO and MGCO (0.3 g·L$^{-1}$) at pH 5.0 for 30 min for Th(IV) and for MB on GCO, 90 min for Pb(II) on GCO and MGCO and 45 min for MB on MGCO at room temperature.

| Name of Adsorbent | Pollutant | Langmuir | | | | Freundlich | | | | Temkin | | | |
|---|---|---|---|---|---|---|---|---|---|---|---|---|---|
| | | $q_{max}$, mg·g$^{-1}$ | $K_L$, L mg$^{-1}$ | $R^2$ | $\chi^2$ | $K_F$, mg·g$^{-1}$ (L·mg$^{-1}$)$^{1/n}$ | $n$ | $R^2$ | $\chi^2$ | $B$, g·L$^{-1}$ | $K_T$, L·mg$^{-1}$ | $R^2$ | $\chi^2$ |
| GCO | Th(IV) | 52.63 | 0.73 | 0.991 | 25.69 | 20.47 | 2.20 | 0.924 | 45.69 | 40.49 | 15.59 | 0.966 | 32.65 |
| | Pb(II) | 47.39 | 0.86 | 0.993 | 19.65 | 16.95 | 2.23 | 0.954 | 52.65 | 20.11 | 11.63 | 0.952 | 42.12 |
| | MB | 111.12 | 1.10 | 0.997 | 22.36 | 52.48 | 2.67 | 0.916 | 39.65 | 20.60 | 120.60 | 0.914 | 40.25 |
| MGCO | Th(IV) | 76.02 | 0.89 | 0.991 | 18.56 | 30.88 | 1.67 | 0.861 | 38.69 | 22.15 | 14.30 | 0.925 | 35.64 |
| | Pb(II) | 71.94 | 0.92 | 0.993 | 28.96 | 27.45 | 1.73 | 0.830 | 42.58 | 32.31 | 12.63 | 0.912 | 40.25 |
| | MB | 76.92 | 0.30 | 0.991 | 22.36 | 47.02 | 4.03 | 0.890 | 36.65 | 32.37 | 65.80 | 0.882 | 38.79 |

The statistical error, chi-square ($\chi^2$), and the linear correlation coefficient ($R^2$) have been used to evaluate the suitability of the isotherm model. The $\chi^2$ value was calculated using the following equation:

$$\chi^2 = \sum \frac{\left(q_{exp} - q_{model}\right)^2}{q_{model}} \tag{6}$$

here $q_{exp}$ and $q_{model}$ are the adsorption capacities obtained from the experimental data and predicted from the model equation, respectively in mg·g$^{-1}$. The $\chi^2$ values observed from Table 2 for the three isotherms followed in the order: Langmuir < Freundlich < Temkin. Hence, the biosorption data for Th(IV), Pb(II), and MB fit better to the Langmuir model, as the $\chi^2$ value was lower than that of the other two models tested here. This suggests that the adsorptive removal of Th(IV), Pb(II), and MB was monolayer adsorption on the homogeneous surface of GCO and MGCO. The maximum adsorption capacity ($q_{max}$) for Th(IV), Pb(II), and MB was found to be 52.63, 47.39, and 111.12 mg·g$^{-1}$ for GCO, and 76.02, 71.94, and 76.92 mg·g$^{-1}$ for MGCO. The adsorption capacity of MGCO for MB was lower than that of GCO and higher than that of graphitic carbon (GC); however, MGCO can be recycled and reused by applying an external magnetic field. The adsorption removal percentage of Th(IV) and Pb(II) was enhanced with the magnetization of GCO, whereas for MB, the trend was opposite.

### 3.4. Reusability of MGCO

Separation and reusability is a paramount analysis for the adsorbent application in wastewater treatment. The adsorption efficacy of MGCO with repeated usage was investigated. MGCO was magnetically separated from the dispersion after the adsorptive removal of each pollutant and was reused for the adsorption of Th(IV), Pb(II), and MB in solution after desorption of each pollutant at each cycle as discussed above. MGCO was again magnetically separated and the experiment was repeated. The adsorption results for MGCO are shown in Figure 5. Up to 5 cycles, MGCO retained 70% efficiency for Th(IV), Pb(II), and MB adsorption.

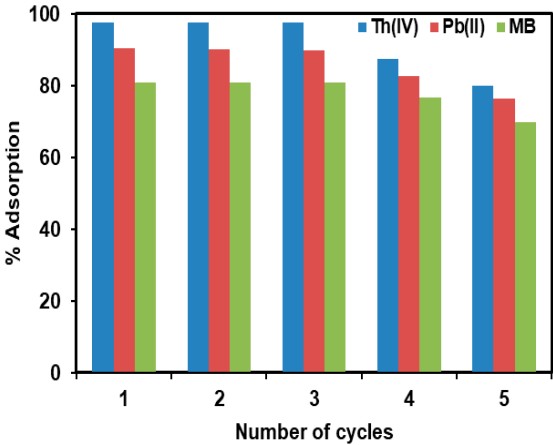

**Figure 5.** MGCO reusability test for adsorptive removal of Th(IV), Pb(II), and MB from water.

### 3.5. Characterization of Texture and Surface of Adsorbents and Adsorption Mechanism

Figure 6 indicates the FT–IR spectra for the GC, GCO, and MGCO. The peaks at 3761 and 3115 cm$^{-1}$ are ascribed to –OH groups and those at 1699 and 1713 cm$^{-1}$ indicate –C=O groups on GCO and MGCO. For MGCO, the –OH peak was slightly shifted to lower wavenumber (3115 cm$^{-1}$).

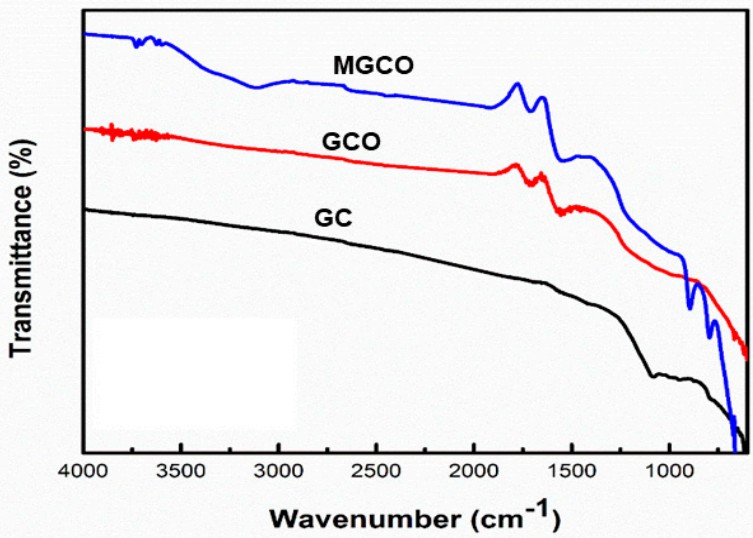

**Figure 6.** FT−IR spectrum of GC, GCO, and MGCO to evaluate surface functional groups.

The data indicate that the –OH group is involved in MGCO formation. The peak positioned at 612 cm$^{-1}$ indicates Fe–O stretching vibration of MGCO composite. This confirms that MGCO was coated with magnetite (iron oxide). X-ray powder diffraction (XRD) analysis of GCO and MGCO (Figure 7a) shows a peak at 2θ = 26.5°, corresponding to crystalline graphitic carbon. For GCO, the primary diffraction peak at 2θ = 24.4°, along with the peak around at 10°, indicates graphitic carbon oxide, and the other smaller and broader peaks indicate amorphous carbon. Thus, along with graphitic carbon oxide, amorphous carbon impurities were present. The XRD pattern of MGCO (Figure 7a) shows peaks at 2θ = 30°, 35°, 43°, 54°, 57°, and 63°, matching the standard peaks of magnetite (JCPDS No. 96-900-2321). It also shows a small peak around at 2θ = 8°, along with the aforementioned magnetite peaks, indicating the formation of MGCO. Moreover, the XRD pattern indicates the crystalline nature of the materials. The Raman spectrum of GCO (Figure 7b) shows two peaks at 1358 and 1596 cm$^{-1}$, which are deployed to the D and G bands, respectively. It is fact, the D and G bands are related to the vibration of disordered sp$^3$ carbon atoms and sp$^2$ carbon atoms of the material, respectively [31]. Upon magnetization of GCO (i.e., the formation of MGCO), the D and G bands shifted to higher wavenumber (1366 and 1604 cm$^{-1}$, respectively), indicative of disordering of the sp$^3$ and sp$^2$ carbons of GCO during the formation MGCO.

The surface characteristics of GCO and MGCO were evaluated using nitrogen adsorption-desorption isotherms, and the measured parameters are shown in Table 3. The Brunauer-Emmett-Teller (BET) surface area of GCO and MGCO was found to be 655.894 m$^2 \cdot$g$^{-1}$ and 428.382 m$^2 \cdot$g$^{-1}$, respectively, indicating that loading of Fe$_3$O$_4$ onto GCO decreased the surface area. This means that iron oxide may have been loaded into the pores of the material. This was confirmed by analysis of the pore volumes of the prepared materials, where the pore volume of MGCO was lower than that of GCO, with an increase in the pore diameter, implying the loading of iron oxide into the pores of GCO. The pore volume of GC and GCO was not significantly different. From the pore volume and diameters, it is concluded that the prepared materials are mesoporous crystalline materials.

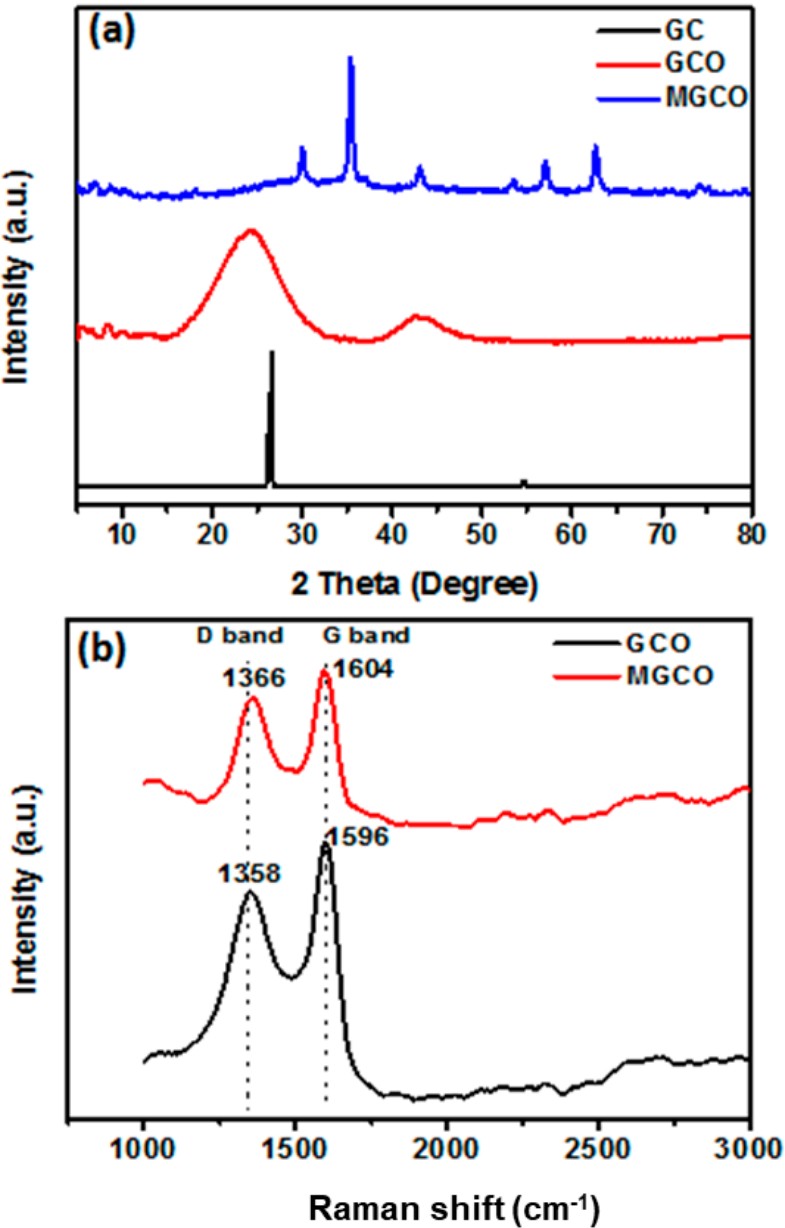

**Figure 7.** XRD (**a**) and Raman (**b**) spectra of GCO and MGCO for identification and confirmation of structure.

**Table 3.** Surface properties of GC, GCO, and MGCO.

| Parameters | GC | GCO | MGCO |
|---|---|---|---|
| BET Surface Area ($m^2 \cdot g^{-1}$) | 674.593 | 655.894 | 428.382 |
| Langmuir Surface Area ($m^2 \cdot g^{-1}$) | 746.798 | 728.914 | 506.278 |
| Pore volume ($cm^3 \cdot g^{-1}$) | 0.278 | 0.259 | 0.146 |
| Average pore diameter (nm) | 1.647 | 2.181 | 5.223 |

The morphology of GCO and MGCO were fully analyzed by TEM and FE-SEM, as shown in Figure 8. Figure 8a,b shows the TEM images of GCO and MGCO, respectively. The TEM image of MGCO clearly shows the $Fe_3O_4$ particles loaded on GCO with proper spherical shapes, compared with GCO. Moreover, the observed MGCO particles aggregated due to iron oxide, thus decreasing the surface area and pore volume of MGCO. The average particle diameter of dispersed $Fe_3O_4$ particles

on the surface of GCO was found to be 16 nm from TEM images. Figure 8c,d shows the SEM image and Figure 8e,f shows the SEM-EDX data; Figure 8g,h presents the FE-SEM data for GCO and MGCO, respectively. From the SEM-EDX analysis, GCO contained 75.54% carbon and 24.46% O and MGCO comprised 70.58% carbon, 19.43% O, and 9.25% Fe. The presence of Fe in MGCO suggests the magnetization of the compound. Further, the magnetization of MGCO was confirmed by using a magnetic properties measurement system (MPMS) having a superconducting quantum interference device (SQUID) magnetometer. MPMS analysis of MGCO found a saturation magnetization ($M_S$) of 41.38 emu·g$^{-1}$ with small remanence ($M_R$ = 3.5 emu·g$^{-1}$) and coercivity ($H_C$ = 52 Oe) at 300 K, indicating that the prepared MGCO was superparamagnetic. This *Ms* is enough for separation of the sorbent from the reaction medium by applying an external magnetic field.

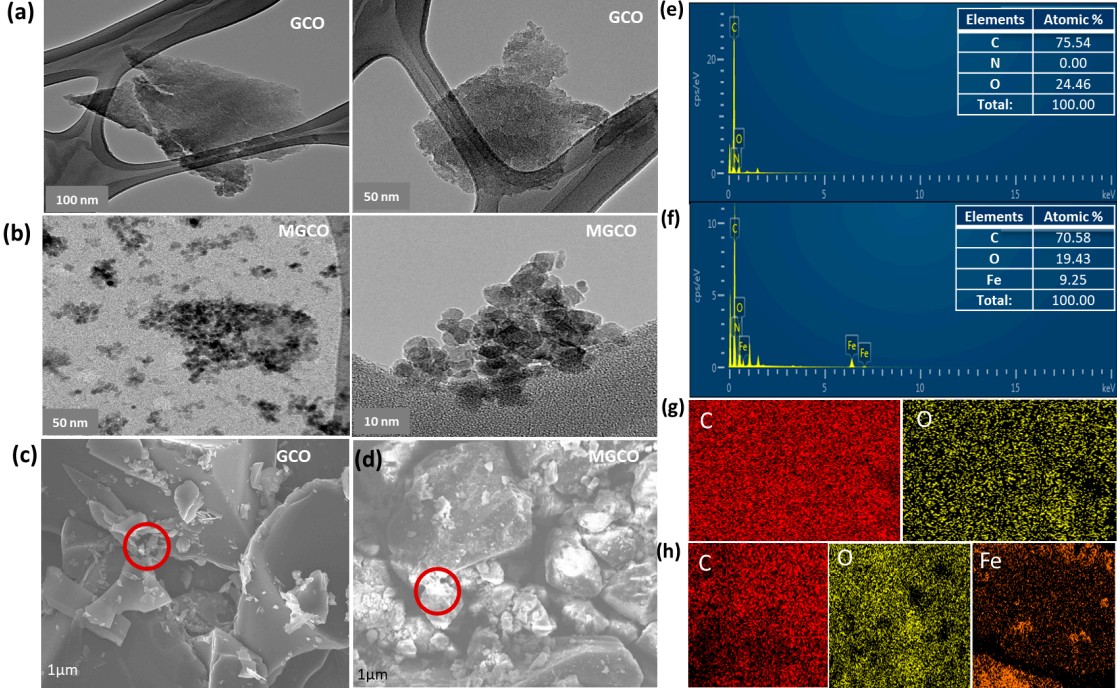

**Figure 8.** TEM image (**a**,**b**), SEM image (**c**,**d**), SEM-EDX (**e**,**f**), and FE-SEM image of GCO (**g**) and MGCO (**h**) to evaluate surface morphology and chemical and elemental texture. The red circled area (**c** and **d**) was used for SEM-EDX analysis of each case (**e** and **f**).

The thermal gravimetric analysis (TGA) results for GCO and MGCO are shown in Figure 9a for evaluation of their stability. In the initial heating stage up to 80 °C, GCO underwent a 12% weight loss due to the loss of the water from the sample. Decomposition started around 500–600 °C, where the mass loss (85%) corresponded to the decay of oxygen-having functional groups in GCO to carbon dioxide. While MGCO displayed 8% weight loss from 25 to 80 °C (8 wt. %) due to dehydration of surface-adsorbed water molecules. An insignificant loss in mass of MGCO was observed between 80 to 370 °C. This insignificant loss in mass of MGCO may be due to the slightly oxidize of $Fe^{2+}$ ($Fe_3O_4$) to $Fe^{3+}$ ($Fe_2O_3$). Even though the MGCO at 370 °C having enough magnetic strength to recovery from aqueous solution by an external magnetic field (which was checked in our laboratory). Again, weight loss occurred between 370 and 600 °C (42 wt. %), possibly due to the decomposition and vaporization of various functional groups at different positions on MGCO. A 50% mass loss of MGCO, along with 50% residual mass at 800 °C, was clearly observed [32]. This indicates that MGCO was stable up to 370 °C after removal of water molecules and insignificant loss in mass, whereas GCO was stable up to 500 °C.

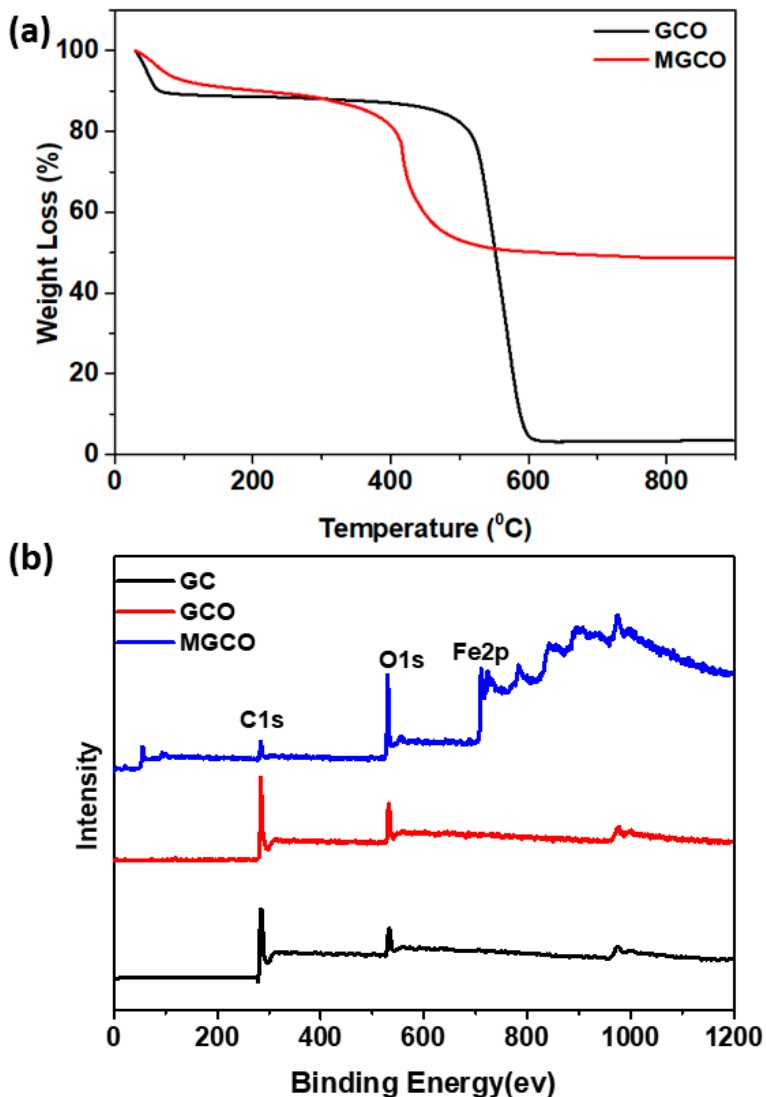

**Figure 9.** TGA analysis of GCO and MGCO for evaluation of their thermal stability and (**a**) and XPS of GC, GCO, and MGCO (**b**) to understand the surface elemental composition.

The XPS spectrum of MGCO composite (Figure 9b) shows the binding energy peaks at 713 and 728 eV correspond to Fe $2p_{3/2}$ and Fe $2p_{1/2}$, respectively, consistent with previous literature [33]. From this result, the iron oxide (Fe(II) and Fe(III)) deposited on GCO was identified as predominantly the magnetite ($Fe_3O_4$) phase. The XPS peak at 284.5 eV indicates C–C for GCO and MGCO. For GC, C–OH and C–O bonds were indicated by peaks at 286.2 and 287.2 eV. Compared with those of GCO, the peaks of MGCO shifted slightly to higher eV values. When compared with GC, GCO and MGCO contain carbonyl and carboxylic groups. A comparison of the binding energy of carbon in MGCO before and after Th(IV), Pb(II), and MB adsorption is shown in Figure 10. The peaks at 286, 287.9, and 289.1 eV for carbon shifted to 286.9, 288, and 289.3 eV for Th(IV), 285.4, 288.2, and 289.3 eV for Pb(II), and 285.2, 288.1, and 289.2 eV for MB respectively. This may be attributed to facile migration of the C atoms to Th(IV), Pb(II), and MB. Therefore, the XPS spectra provided evidence of the formation of complexes between Th(IV), Pb(II), and MB and the carbon atoms of MGCO.

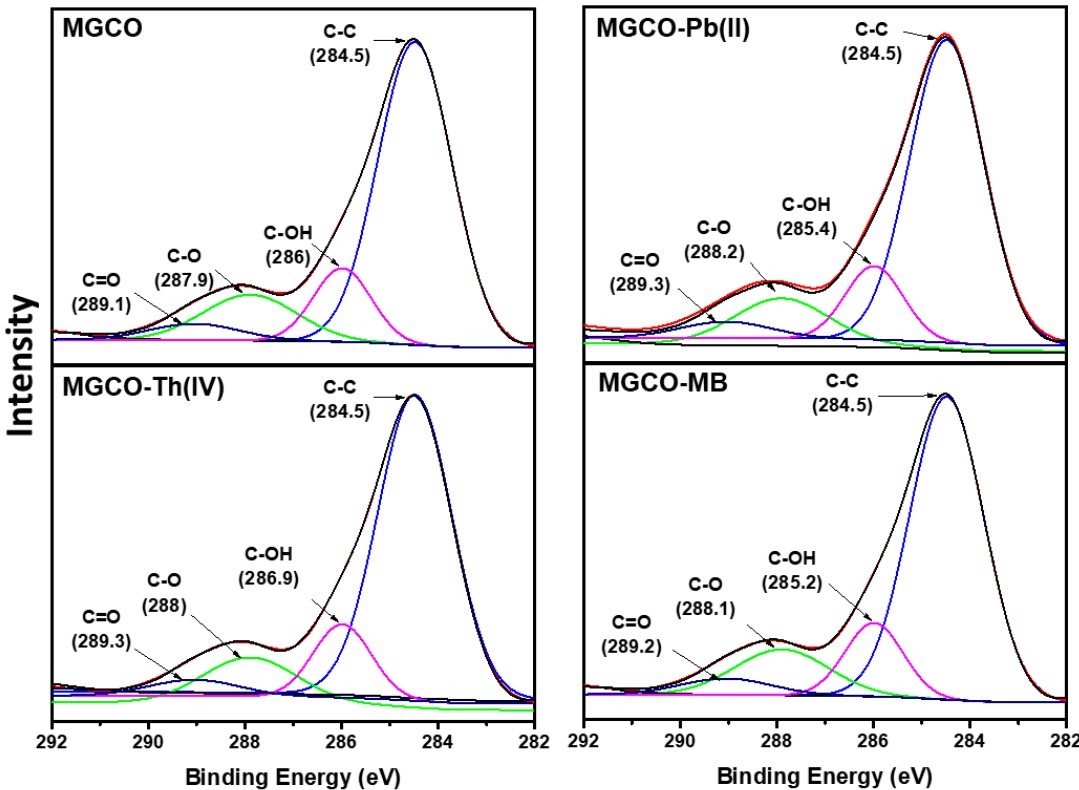

**Figure 10.** High-resolution XPS C1s spectrum of MGCO of before and after Th(IV), Pb(II), and MB adsorption to evaluate adsorption mechanism.

## 4. Conclusions

In the present study, GCO and MGCO were prepared from edible sugar and were used for the adsorptive deposition of Th(IV), Pb(II), and MB from wastewater. The green ozone oxidation of graphitic carbon was well optimized. Prior to the test the adsorption ability of prepared GCO and MGCO, the physical and chemical properties, including texture and surface morphology, were well characterized by instrumental analytical techniques. The MPMS studies confirmed the superparamagnetism of MGCO having enough saturated magnetization, 41.38 emu·g$^{-1}$ for the recovery of material from water. Moreover, both GCO and MGCO are thermally stable up to 400 °C was confirmed from TGA. Further, the adsorption process was well predicted with the aid of high-resolution XPS spectrum, SEM-EDX, and FE-SEM that have confirmed the adsorption process is mainly due to the carboxylic and epoxy or ether, and Fe–O surface functional groups interaction with pollutants. The batch kinetics shows that both metal ions and dye removal by GCO and MGCO are the rate-limiting reactions. Isothermal analysis indicates that the adsorption capacity of GC for Th(IV), Pb(II), and MB was 52.63, 47.39, and 111.12 mg·g$^{-1}$, respectively, with corresponding values of 76.02, 71.94, and 76.92 mg·g$^{-1}$ for MGCO. The adsorptive removal capacity of GCO and MGCO for Th(IV), Pb(II) and MB is comparable with that of other reported materials. The present prepared MGCO can be reusable up to 3 cycles without loss of its ability. From overall results, it is concluded that GCO and MGCO are highly efficient and potential materials for the adsorptive recovery of Th(IV), Pb(II), and MB from water. Cost analysis of the preparation of GCO and MGCO is expected to show that they are economical compared with activated carbon materials.

**Supplementary Materials:** The following materials are available online at http://www.mdpi.com/2075-4701/9/9/935/s1 for review: Materials and methods, including 2.1. Materials, and 2.2. Analytical methods; Figure S1: Adsorption kinetic models of Th(IV), Pb(II), and MB on to GCO and MGCO.; Figure S2. Salt effect of Th(IV), Pb(II), and MB on to GCO and MGCO.; Figure S3. Adsorption isotherms models of Th(IV), Pb(II), and MB on to GCO and MGCO.

**Author Contributions:** L.P.L., J.R.K., Y.-Y.C. conceived and designed the experiments; L.P.L., Y.-L.C., J.-S.C. contributed to conducting experiments; J.-K.Y. contributed analysis tools; L.P.L., J.R.K. analyzed the data, writing-review & editing.

**Funding:** "This research was funded by the National Research Foundation (NRF) of Korea, funded by the Ministry of Science, ICT & Future Planning (MSIP), grant number 2017R1C1B5016656" and "The APC was funded by the Kwangwoon University, Seoul, Korea through Research Grant–2019 ".

**Acknowledgments:** This research was financially contributed by the National Research Foundation (NRF) of Korea, funded by the Ministry of Science, ICT & Future Planning (MSIP) (2017R1C1B5016656) of the Korean Government. It was also partially supported by the Kwangwoon University, Seoul, Korea through Research Grant–2019.

**Conflicts of Interest:** The authors declare no conflict of interest between authors or any institute or funding agencies.

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
