# Peer review of "Green Activated Magnetic Graphitic Carbon Oxide and Its Application for Hazardous Water Pollutants Removal"

_metals, doi:10.3390/met9090935_

Round 1

Reviewer 1 Report

Dear Editor,

The manuscript entitled "Green activated magnetic graphitic carbon oxide and its application for hazardous water pollutants removal" presents a comparative study of the ability of graphitic carbon oxide and magnetic graphitic carbon oxide to adsorb Th(IV), Pb(II) and methylene blue from wastewater. The conditions of synthesis of magnetic graphitic carbon oxide are also presented. The authors have used appropriate methods of characrerizing the adsorption process. I have some remarks and questions to the authors

There is a typo on line 91. "for amking" should be "for making". The adsorption of Th(IV), Pb(II) and methylene blue (Fig. 2) on graphitic carbon oxide is equal. How do you interpret this? The table 1 is spread over two pages. Please, do not split it. The scale bar on the SEM/TEM images (Fig.7) is not visible. It is impossible to find out what the resolution is. The Fe3O4 particles have usually a cubic shape or a spherical one for very small particles. In general, magnetite can't grow to a hexagonal shape. Is it possible that you deal with other iron oxides or other materials? Can you identify which particles on Fig. 7 have a hexagonal shape? Please, indicate the where area the EDX analysis was conducted. How were the samples prepared for it? What is the average particle size of Fe3O4? You indicate that the magnetic graphitic carbon oxide is stable up to 370 °C. But Fe3O4 is oxidized at temperatures >100°C to Fe2O3, a material with a low Ms. Did you observe any changes in Ms and is the value of magnetization sufficient for magnetic separation?

Sincerely yours

Author Response

Reviewer 1

The manuscript entitled "Green activated magnetic graphitic carbon oxide and its application for hazardous water pollutants removal" presents a comparative study of the ability of graphitic carbon oxide and magnetic graphitic carbon oxide to adsorb Th(IV), Pb(II) and methylene blue from wastewater. The conditions of synthesis of magnetic graphitic carbon oxide are also presented. The authors have used appropriate methods of characterizing the adsorption process. I have some remarks and questions to the authors. 

Answer:

We appreciate the valuable comments of all reviewers on our manuscript. The followings are the explanations presented in reply to each reviewers’ comment. The critical comments and useful suggestions have been helped us to improve our paper considerably. As indicated in the reply’s that follow, we have taken these comments and suggestions into account in the revised version of our manuscript and marked with text with yellow background color in the revised manuscript.

There is a typo on line 91. "for amking" should be "for making".

Answer: We rectified it.

The adsorption of Th(IV), Pb(II) and methylene blue (Fig. 2) on graphitic carbon oxide is equal. How do you interpret this?

Answer: Here we used different dosages of GCO to get equal pollutant removal for each case of pollutant, “The GCO used for each case are 0.8g L-1 for Th(IV), 0.9g L-1 for Pb(II) and 0.3g L-1 for MB” based adsorbent dosage effect. That is the reason here we get the same adsorption removal percentages for all pollutants. To avoid the confusion to the readers, now we did the experiment by using 1 gL-1 of GCO prepared from ozonation process for the adsorption performance of three pollutants and was reported here instead of old ones in the revised manuscript. This clearly shows in the difference adsorption removal for different pollutants. For your perusal see Figure 2.

Figure 2. Adsorption performance of GCO (1.0 g L-1) for three pollutants (Th(IV), Pb(II) and MB = 10 mg L-1) prepared from ozonation of GC at different pH (2, 4, 6, 8, 10, and 12), temperature 25 °C, and time 1 h.

Table 1 is spread over two pages. Please, do not split it.

Answer: We rectified it.

The scale bar on the SEM/TEM images (Fig.7) is not visible. It is impossible to find out what the resolution is. The Fe3O4 particles have usually a cubic shape or a spherical one for very small particles. In general, magnetite can't grow to a hexagonal shape. Is it possible that you deal with other iron oxides or other materials? Can you identify which particles in Fig. 7 have a hexagonal shape? Please, indicate where the area the EDX analysis was conducted. How were the samples prepared for it?

Answer: We agree with you, and thank you for the good question. We indicate the scale at Fig.7. The shape of particles is spherical not in hexagonal the same was rectified in the text. The area used for EDX was indicated at SEM images (c and d) with a red circle. We prepared samples for SEM analysis by dispersing the grounded powdered material on carbon tip, which is inserted for SEM analysis. It is a typical method as often used for SEM analysis.

What is the average particle size of Fe3O4? You indicate that the magnetic graphitic carbon oxide is stable up to 370 °C. But Fe3O4 is oxidized at temperatures >100°C to Fe2O3, a material with a low Ms. Did you observe any changes in Ms and is the value of magnetization sufficient for magnetic separation?

Answer: The average particles diameter of Fe3O4 found to be 16 nm measured from TEM and the same included in the main manuscript lines 336-337 on page 12.

We checked Ms of MGCO at -5oC and 25 oC (Room temperature) only. We did not check at >100 oC. However, a slight insignificant mass loss in MGCO was observed between 80 to 370 oC in TGA. This can be ascribed to the slight oxidases of Fe2+ (Fe3O4) to Fe3+ (Fe2O3). Even though the MGCO at 370 0C having enough magnetic strength to recovery from aqueous solution by an external magnetic field (which was we checked in a laboratory). Moreover, in TGA analysis, we checked only the stability of material that found to the material stable up to 370 oC after removal of water molecules (> 80 oC) and insignificant loss in mass after >100 oC.

Reviewer 2 Report

August 2019

Metals

Title:  Green Activated Magnetic Graphitic Carbon Oxide and its Application for Hazardous Water Pollutants Removal

By: Lakshmi Prasanna Lingamdinne, Jong-Soo Choi, Yu-Lim Choi, Jae-Kyu Yang*, Janardhan Reddy Koduru*, Yoon-Young Chang*

Department of Environmental Engineering, Kwangwoon University, Seoul, 01897, Republic of Korea

Review on Manuscript Number: Metals-578643

Comments for the manuscript

Summary

This paper presents a study about the preparation of cost-effective graphitic carbon oxide (GCO) and magnetic graphitic carbon oxide (MGCO) from edible sugar, via optimized green activation and employing ozone oxidation. The aim of the study was to assess the characteristics of the prepared materials for adsorption of water pollutants, including organic pollutants, metals, and radionuclides. During the study, the authors evaluate the surface adsorptive performance of prepared GCO and MGCO for Th(IV), Pb(II) and MB adsorptive removal.

Overall Opinion

As it is mentioned by the authors, water pollution is a primary universal environmental issue, and the release of harmful pollutants into water bodies is well above the standard limits. Therefore, in one hand, it is important to improve mechanisms of water pollutants detection. On the other hand, owing to the diversity of harmful products circulating through the environment, it is crucial to improve the current knowledge about methodologies to control and/or remediate polluted water. This study provides important information about the adsorption of Th(IV), Pb(II) and MB on GCO and MGCO, and the possible use of these materials in the control of water quality.

In this context, I consider that the type of research work presented and discussed in the manuscript meets the publishing objectives of Metals and would be adequate to be published in this scientific journal.

The presentation of data, methodology and discussion is adequately founded and illustrated with appropriate tables and figures. Supplementary material for on-line publication is adequate and complements the text of the manuscript.

However, in my opinion the manuscript needs a revision before publication, mainly related with parts of the text that need to be clarified, the English grammar and typos in the text.

General comments

Self-citation - In my opinion, the citation of previous works by the authors in general statements is not appropriate. Examples are the sentences in pages 1-2 (lines 34 to 37) with citations [1] and [2]. The comments are of current knowledge and the cited studies from the same authors do not add information.

In Section 1. Introduction (line 77) consider to use the verbs in the past tense (the main objectives of this study were…)

Minor suggestions

The titles of different sub-sections should be harmonised:

2.1. Graphitic carbon preparation

2.2. Preparation of graphitic carbon oxide (GCO)

2.3. Preparation of magnetic graphitic carbon oxide (MGCO)

Page 2, line 83 – “…previous papers…”

Page 3, line 91 – “…temperature for amking…” making?”

Section 2.3. needs revision to correct typos in the text and clarify and/or correct some of the sentences.

Page 3, line 102 – “As showin in Fig.1, to the above prepared 1 g of GCO has been dispersed in 500 mL water…” show in; remove the word to.

Page 3, line 104 – “stirring and inert or (N2) atmosphere.” …on?

Page 3, line 105 – “The solution warmed up to 80 °C and stirred for 45 min.”- instead “The solution was warmed up to 80 °C and stirred for 45 min.”

Page 3, lines 107-108 – “The material has been collected with aid of external magnet, water washed, and dried at 80 °C in a vacuum oven for 8 h.” instead “…with aid of an external magnet, washed with water and dried at 80 °C in a vacuum oven for 8 h.”

Page 3, lines 108-109 – “The prepared material physicochemical properties was confirmed by X-ray diffraction (XRD),…” instead “The physicochemical properties of the prepared material was confirmed by X-ray diffraction (XRD),…”

Page 4, lines 115-116 – “Figure 1. Schematic presentation of green oxidation of graphitic carbon using O3 is produced with aid of ozone generator for the production of GCO and MGCO.” …is produced?

Page 4, line 120 – No previous mention to MB (only in the supplementary material)

Page 4, lines 125-126 – “The filtrate contain pollutants concentration was measured by using inductively coupled plasma-optical emission spectroscopy… to estimate the adsorptive amount of pollutant” – the sentence needs English revision.

Page 5, lines 144 and 150 – “…are indicates…”?

Section 3. “Results and discussions” instead “Results and discussion”

Page 5, lines 162 and 163 – “…the adsorptive removal of the pollutants declined by the solution pH increases from 6 to 12…” - “…declined by…”- the sentence needs English revision.

Page 5, lines 166 and 168 – “The greater number of oxygen-containing functional groups in GCO prepared by ozonation at lower pH, as compared to that at higher pH, the GCO prepared at lower pH shows higher adsorption.” – this sentence needs to be clarified, as it is written is not clear.

Page 5, lines 168 and 170 – “The other experimental parameters used in the ozone experiment were: temperature 25 °C, time 1 h, and a disc-type bubbler were used in the ozonation process.” – the same comment about the structure of the sentence, as the previous.

Section “3.2. Experimental parameters for adsorptive removal of pollutants by MGCO and GCO” – in the text is mentioned first Fig. 3 and then Table 1. The order of appearance of these should follow the order indicated in the text.

Table 1 - Column 2 – I suppose is methylene blue and it is not a metal ion. Consider use only the term ion.

Page 6, lines 186 – “Fig. 3(a,b) found that…” – Figure - at the beginning of the sentence; figure found that? consider instead “Figure 3 shows that…”.

Page 7, lines 203-204 – “The removal percentage of GCO and MGCO for Th(IV), Pb(II), and MB at different initial pH of the solution are illustrated in Fig. 3(c,d).” – removal percentages… are illustrated (verb).

Section “3.3. Adsorption isotherms” – in the text is mentioned first Fig. 4 and then Table 2. The order of appearance of these should follow the order indicated in the text.

Page 8, line 242 – “…deferent…” - different?

Page 9 – Table 2, Column 2 – I suppose is methylene blue and it is not a metal ion. Consider use only the term ion.

Page 10, line 274 – “…paraamount…” – paramount?

Page 11, line 287 – Fig.S4 should be included in the manuscript, because an interpretation of the data is presented.

Page 11, line 291 – “…ascribed…” – ascribed?

Page 10, lines 301-305 – please confirm if the statement in this sentence is in accordance with the data shown in Fig. 6b. Are the colours in the graph drawn? (“The Raman spectrum of GCO (Fig. 6(b)) shows two peaks at 1366 and 1604 cm–1… Upon magnetization of GCO (i.e., formation of MGCO), the D and G bands shifted to lower wavenumber (1358 and 1596 cm–1, respectively)…”- in the Figure it seems the opposite; the text is not clear).

Page 14, line 349 – “This can be ascribed to the evaporation of.” The sentence is not complete…

Supplementary material

Figure S2 – the a), b)… to f) report to which conditions mentioned in the legend of the figure?

Figure S3 – the same comment as previous

Author Response

Reviewer 2

Comments for the manuscript

Summary

This paper presents a study about the preparation of cost-effective graphitic carbon oxide (GCO) and magnetic graphitic carbon oxide (MGCO) from edible sugar, via optimized green activation and employing ozone oxidation. The aim of the study was to assess the characteristics of the prepared materials for adsorption of water pollutants, including organic pollutants, metals, and radionuclides. During the study, the authors evaluate the surface adsorptive performance of prepared GCO and MGCO for Th(IV), Pb(II) and MB adsorptive removal.

Overall Opinion

As it is mentioned by the authors, water pollution is a primary universal environmental issue, and the release of harmful pollutants into water bodies is well above the standard limits. Therefore, on one hand, it is important to improve mechanisms of water pollutants detection. On the other hand, owing to the diversity of harmful products circulating through the environment, it is crucial to improve the current knowledge about methodologies to control and/or remediate polluted water. This study provides important information about the adsorption of Th(IV), Pb(II) and MB on GCO and MGCO, and the possible use of these materials in the control of water quality.

In this context, I consider that the type of research work presented and discussed in the manuscript meets the publishing objectives of Metals and would be adequate to be published in this scientific journal.

The presentation of data, methodology, and discussion is adequately founded and illustrated with appropriate tables and figures. Supplementary material for on-line publication is adequate and complements the text of the manuscript.

However, in my opinion, the manuscript needs a revision before publication, mainly related to parts of the text that need to be clarified, the English grammar and typos in the text.

Answer: We are very much thankful to you for your comments and spending your valuable time to review our manuscript. The critical comments and useful suggestions have been helped us to improve our paper considerably. We have taken these comments and suggestions into account in the revised version of our manuscript and marked with text with yellow background color in the revised manuscript.

General comments

Self-citation - In my opinion, the citation of previous works by the authors in general statements is not appropriate. Examples are the sentences in pages 1-2 (lines 34 to 37) with citations [1] and [2]. The comments are of current knowledge and the cited studies from the same authors do not add information.

Answer: Thank you for the good advice; we added some other author’s references for that sentence.

In Section 1. Introduction (line 77) consider using the verbs in the past tense (the main objectives of this study were…)

Answer: we rectified it.

Minor suggestions

The titles of different sub-sections should be harmonised:

2.1. Graphitic carbon preparation

2.2. Preparation of graphitic carbon oxide (GCO)

2.3. Preparation of magnetic graphitic carbon oxide (MGCO)

Answer: We harmonized those titles of Section 2.1, 2.1 and 2.3.

Page 2, line 83 – “…previous papers…”

Answer: We rectified it

 Page 3, line 91 – “…temperature for amking…” making?”

Answer: We rectified it.

Section 2.3. Needs revision to correct typos in the text and clarify and/or correct some of the sentences.

Answer: Thank for the good suggestions, we revised and rectified it as you suggested.

Page 3, line 102 – “As showin in Fig.1, to the above prepared 1 g of GCO has been dispersed in 500 mL water…” show in; remove the word to.

Answer: We rectified it

Page 3, line 104 – “stirring and inert or (N2) atmosphere.” …on?

Answer: We rectified it

Page 3, line 105 – “The solution warmed up to 80 °C and stirred for 45 min.”- instead “The solution was warmed up to 80 °C and stirred for 45 min.”

Page 3, lines 107-108 – “The material has been collected with aid of external magnet, water washed, and dried at 80 °C in a vacuum oven for 8 h.” instead “…with aid of an external magnet, washed with water and dried at 80 °C in a vacuum oven for 8 h.”

Page 3, lines 108-109 – “The prepared material physicochemical properties was confirmed by X-ray diffraction (XRD),…” instead “The physicochemical properties of the prepared material was confirmed by X-ray diffraction (XRD),…”

Answer: We revised the all above comments in section 2.3 and rectified as per you suggested.

Page 4, lines 115-116 – “Figure 1. Schematic presentation of green oxidation of graphitic carbon using O3 is produced with aid of ozone generator for the production of GCO and MGCO.” …is produced?

Answer: We rectified it.

Page 4, line 120 – No previous mention to MB (only in the supplementary material)

Answer: We added MB in abstract and Introduction at lines 37 and 44.

Page 4, lines 125-126 – “The filtrate contain pollutants concentration was measured by using inductively coupled plasma-optical emission spectroscopy… to estimate the adsorptive amount of pollutant” – the sentence needs English revision.

Answer: We revised that sentence at lines 126-129 in the revised manuscript.

Page 5, lines 144 and 150 – “…are indicates…”?

Answer: we revised it in the revised manuscript at Line 145 and 151.

Section 3. “Results and discussions” instead “Results and discussion”

Answer: We rectified it.

Page 5, lines 162 and 163 – “…the adsorptive removal of the pollutants declined by the solution pH increases from 6 to 12…” - “…declined by…”- the sentence needs English revision.

Answer: We rectified it. See page 5 lines 162 and 163 as changed to “the adsorptive removal of the pollutants decreases to lower range by the solution pH increases…”

Page 5, lines 166 and 168 – “The greater number of oxygen-containing functional groups in GCO prepared by ozonation at lower pH, as compared to that at higher pH, the GCO prepared at lower pH shows higher adsorption.” – this sentence needs to be clarified, as it is written is not clear.

Answer: We revised that sentence as “The greater number of oxygen-containing functional groups has been observed in GCO prepared by ozonation carried out at lower pH as compared to that ozonation carried out at higher pH from FT-IR (it is not reported here). Due to the high oxygen–containing functional groups of the GCO prepared at lower pH shows higher adsorption.” In page 5, Lines 166-171 in the revised manuscript.

Page 5, lines 168 and 170 – “The other experimental parameters used in the ozone experiment were: temperature 25 °C, time 1 h, and a disc-type bubbler were used in the ozonation process.” – the same comment about the structure of the sentence, as the previous.

Answer: We revised as “Further, ozonation time effect on GC oxidation was studied and the results found that the ozonation time was an insignificant effect on GC oxidation. From this optimization studies, pH 4.0, temperature 25 °C, time 1 h, and a disc-type bubbler were used in the ozonation process as an optimized condition for the oxidation of GC to GCO.” In Page5, Lines170 to 173 in the revised manuscript.

Section “3.2. Experimental parameters for adsorptive removal of pollutants by MGCO and GCO” – in the text is mentioned first Fig. 3 and then Table 1. The order of appearance of these should follow the order indicated in the text.

Answer: We re-ordered them as you suggested in the revised manuscript.

Table 1 - Column 2 – I suppose is methylene blue and it is not a metal ion. Consider using only the term ion.

Answer: we rectified it as a common name “pollutant”

Page 6, lines 186 – “Fig. 3(a,b) found that…” – Figure - at the beginning of the sentence; the figure found that? consider instead “Figure 3 shows that…”.

Answer: We rectified it you suggested inline 190 in the revised manuscript.

Page 7, lines 203-204 – “The removal percentage of GCO and MGCO for Th(IV), Pb(II), and MB at different initial pH of the solution are illustrated in Fig. 3(c,d).” – removal percentages… are illustrated (verb).

Answer: We revised it as “The removal percentages of GCO and MGCO for Th(IV), Pb(II), and MB at different initial pH of the solution are illustrated in Fig. 3(c,d).” at lines 227-228 in page 8.

Section “3.3. Adsorption isotherms” – in the text is mentioned first Fig. 4 and then Table 2. The order of appearance of these should follow the order indicated in the text.

Answer: We rectified it in the revised manuscript.

Page 8, line 242 – “…deferent…” - different?

Answer: we rectified it in the revised version on page 9, line, 257.

Page 9 – Table 2, Column 2 – I suppose is methylene blue and it is not a metal ion. Consider using only the term ion.

Answer: we rectified it as a common name “pollutant” in the revised manuscript.

Page 10, line 274 – “…paraamount…” – paramount?

Answer: We rectified it on page 10, line 288 in the revised manuscript.

Page 11, line 287 – Fig.S4 should be included in the manuscript because an interpretation of the data is presented.

Answer: We included Fig.S4 in the main manuscript as Fig. 6 as you suggest at page 12.

Page 11, line 291 – “…ascribed…” – ascribed?

Answer: We revised that sentence as “The peak positioned at 612 cm−1 indicates Fe–O stretching vibration of MGCO composite.” on page 11, lines 304-305 in the revised manuscript.

Page 10, lines 301-305 – please confirm if the statement in this sentence is in accordance with the data shown in Fig. 6b. Are the colours in the graph drawn? (“The Raman spectrum of GCO (Fig. 6(b)) shows two peaks at 1366 and 1604 cm–1… Upon magnetization of GCO (i.e., the formation of MGCO), the D and G bands shifted to lower wavenumber (1358 and 1596 cm–1, respectively)…”- in the Figure it seems the opposite; the text is not clear).

Answer: It’s a typo mistake. We revised in the text on page 11, lines 315-318 in the revised manuscript.

Page 14, line 349 – “This can be ascribed to the evaporation of.” The sentence is not complete…

Answer: We revised this sentence in the revised manuscript on page 14, lines 365-368 as “An insignificant loss in mass of MGCO was observed between 80 to 370 oC. This insignificant loss in mass of MGCO may be due to the slightly oxidize of Fe2+ (Fe3O4) to Fe3+ (Fe2O3). Even though the MGCO at 370 0C having enough magnetic strength to recovery from aqueous solution by an external magnetic field (which was checked in our laboratory).”

Supplementary material

Figure S2 – the a), b)… to f) report to which conditions mentioned in the legend of the figure?

Answer: we rectified in the revised supplementary.

Figure S3 – the same comment as previous

Answer: we rectified in the revised supplementary.

Round 2

Reviewer 1 Report

Dear Editor,

Authors answer to all of my questions and did the appropriate changes in the manuscript. I recommend the manuscript to be published.

Best regards